# Validity of daily self-pulse palpation for atrial fibrillation screening in patients 65 years and older: A cross-sectional study

**Faris Ghazal** [1] *, **Holger Theobald** [2], **Mårten Rosenqvist** [1], **Faris Al-Khalili** [1]

**1** Department of Clinical Sciences, Danderyd University Hospital, Karolinska Institutet, Stockholm, Sweden, **2** Department of Neurobiology, Care Sciences and Society, Division of Family Medicine and Primary Care, Karolinska Institutet, Stockholm, Sweden

* faris.ghazal@ki.se

## Abstract

### Background

The European Society of Cardiology guidelines recommend (Class IA) single-time–point screening for atrial fibrillation (AF) using pulse palpation. The role of pulse palpation for AF detection has not been validated against electrocardiogram (ECG) recordings. We aimed to study the validity of AF screening using self-pulse palpation compared with an ECG recording conducted at the same time using a handheld ECG 3 times a day for 2 weeks.

### Methods and findings

In this cross-sectional screening study, patients 65 years of age and older attending 4 primary care centers (PCCs) outside Stockholm County were invited to take part in AF screening from July 2017 to December 2018. Patients were included irrespective of their reason for visiting the PCC. Handheld intermittent ECGs 3 times per day were offered to patients without AF for a period of 2 weeks, and patients were instructed in how to take their own pulse at the same time. A total of 1,010 patients (mean age 73 years, 61% female, with an average $CHA_2DS_2$-VASc score 2.9) participated in the study, and 27 (2.7%, 95% CI 1.8%–3.9%) new cases of AF were detected. Anticoagulants (ACs) could be initiated in 26 (96%, 95% CI 81%–100%) of these cases. A total of 53,782 simultaneous ECG recordings and pulse measurements were registered. AF was verified in 311 ECG recordings, of which the pulse was palpated as irregular in 77 recordings (25%, 95% CI 20%–30% sensitivity per measurement occasion). Of the 27 AF cases, 15 cases felt an irregular pulse on at least one occasion (56%, 95% CI 35%–75% sensitivity per individual). 187 individuals without AF felt an irregular pulse on at least one occasion. The specificity per measurement occasion and per individual was (98%, 95% CI 98%–98%) and (81%, 95% CI 78%–83%), respectively.

### Conclusions

AF screening using self-pulse palpation 3 times daily for 2 weeks has lower sensitivity compared with simultaneous intermittent ECG. Thus, it may be better to screen for AF using

**Data Availability Statement:** All relevant data are within the manuscript and its Supporting Information files.

**Funding:** MR received the following fundings for this work 4-1082/2019 Swedish Heart-Lung Foundation, 4-3806/2016 Boehringer Ingelheim, 4.3481/2018 Bayer CropScience, and 4-1804/2015 Pfizer Foundation. All other authors received no specific funding for this work. The funders had no role in study design, data collection and analysis, decision to publish, or preparation of the manuscript.

**Competing interests:** I have read the journal's policy, and the authors of this manuscript have the following competing interests: MR has received research grants, lecture and consulting honoraria from the following sources: Abbott, Carl Bennett AB, Bristol Myers Squibb, Medtronic, MSD, Pfizer, Roche, Sanofi, and Zenicor. FA has received lecture fees from Bristol-Myers-Squibb, Pfizer, Boehringer-Ingelheim, and Bayer. All other authors declared no conflict of interest.

**Abbreviations:** AC, anticoagulant; AF, atrial fibrillation; ECG, electrocardiogram; PCC, primary care center.

intermittent ECG without stepwise screening using pulse palpation. A limitation of this model could be the reduced availability of handheld ECG recorders in primary care centers.

---

## Author summary

### Why was this study done?

- Atrial fibrillation (AF) without anticoagulant (AC) treatment is a risk factor for ischemic stroke. Thus, detection of AF and initiation of ACs may prevent stroke.

- Pulse palpation is recommended for single-time–point screening for AF, which is often paroxysmal and difficult to detect through single-time–point screening.

- Intermittent electrocardiogram (ECG) monitoring over 2 weeks is a sensitive screening method for AF. However, the role of pulse palpation for AF detection has not been validated against simultaneous intermittent ECG recordings.

### What did the researchers do and find?

- Individuals 65 years of age and older without known AF who were seeking care in 4 primary care centers, irrespective of reason, were invited to AF screening using intermittent ECG 3 times a day over 2 weeks and were instructed in how to take their own pulse at the same time.

- A total of 1,010 individuals were screened by intermittent ECG, and 27 (2.7%) new AF cases were detected, although only 5 of these cases were detected at first ECG on inclusion. ACs could be initiated in 26 new AF cases.

- Self-pulse palpation showed a low sensitivity (56%) and high specificity (81%) for AF detection.

### What do these findings mean?

- AF screening using self-pulse palpation daily over 2 weeks is inferior to screening using intermittent ECG.

- It is better to screen for AF using intermittent ECG without pulse palpation.

- Opportunistic screening for AF in primary care is promising because initiation of ACs was high. In future, a randomized control screening study in primary care is needed in order to validate whether the stroke incidence has been reduced.

## Introduction

Atrial fibrillation (AF) without anticoagulant (AC) treatment is a risk factor for ischemic stroke [1]. Stroke may be the first presentation of AF [2]. Silent AF detected by continuous ECG monitoring is associated with an elevated stroke risk [3]. A nonrandomized

interventional screening study for AF and treatment with ACs suggest a reduction in stroke incidence among a screened population compared with an unscreened population [4]. Thus, detection of AF and initiation of ACs may be important for preventing stroke [1]. AF manifests as paroxysmal attacks and can then progress to persistent or permanent AF [5]. Nonparoxysmal AF could be detected through a single-time–point electrocardiogram (ECG), but it is difficult to detect paroxysmal AF [1]. Thus, repeated daily ECG monitoring is probably more sensitive than single-time–point ECG in detecting AF [6].

According to the European Society of Cardiology guidelines 2016 [1], it is recommended that persons 65 years of age and older are screened for AF. This is based on screening studies [7] using single-time–point ECG with or without pulse palpation. These guidelines even consider systematic screening for AF in persons over 75 years of age based on a screening study [6] that used intermittent ECG monitoring over 2 weeks. A consensus paper [8] for AF screening showed differences in the AF detection rate depending on the screening technique and the characteristics of the screened population. However, very few studies have performed a direct comparison between pulse palpation and ECG recordings. Self-detection of AF through pulse palpation is feasible among the elderly, as shown in an interventional study conducted on an anatomical model [9]. Another study showed a high level of motivation in participants to continue taking their pulse over several weeks [10].

Our aim was to study the validity of repeated self-pulse palpation over 2 weeks compared with ECG monitoring to screen for AF among patients 65 years of age and older who were seeking primary healthcare. Finally, we aimed to evaluate the initiation rate of oral ACs for newly detected AF cases to prevent stroke.

## Methods

### Design

A cross-sectional screening study was performed in Swedish primary care centers (PCCs). We planned to recruit 3–6 PCCs that had shown an interest in participating. They were located outside Stockholm County. The reason for this geographical limitation was that another large AF screening study was being conducted within the Stockholm area. The study started in June 2017 in 2 PCCs, and then an additional 2 PCCs were recruited to fulfill the sample size within the planned study period of 2 years. A total of 14 PCCs were approached, and 4 PCCs participated in the study. This study is reported in accordance with the Strengthening the Reporting of Observational Studies in Epidemiology (STROBE) guidelines (S1 Table). The protocol is described on protocols.io (https://www.protocols.io/view/validity-of-daily-self-pulse-palpation-over-two-we-m2ec8be).

### Study population

Patients who, for whatever reasons, were seeking care at a PCC and who were 65 years of age or older were invited by health personnel to participate. Patients who were interested in participating were then directed to a research nurse in the PCC. No incentives were given for participation. Patients with previously known AF or ongoing oral AC treatment were excluded.

### Screening procedure

One to two nurses were assigned per center in this study. The nurses had received prior training in AF and pulse palpation. No incentives or accreditation points were given for this training.

Participants received written and oral information about the study. All participants provided their consent to participate by signing and submitting a consent form before entering the study. In a 30-minute consultation with a nurse, participants completed a questionnaire (Supplement Case Report Form) about stroke risk stratification according to the $CHA_2DS_2$-VASc score. The CHA2DS2-VASc score considers congestive heart failure, hypertension, age, diabetes mellitus, previous stroke/transient ischemic attack, vascular disease, and female sex. Body weight, height, pulse, and blood pressure were measured. The nurse carefully instructed participants in the technique of radial pulse palpation and then checked the participants' ability to perform self-pulse palpation. The participants were instructed to perform self-pulse palpation at home 3 times a day over a 2-week period, immediately followed each time by a 30-second ECG recording using a Zenicor handheld ECG with an integrated mobile transmitter (https://zenicor.se/; Stockholm, Sweden). We did not specify which time of day the participants should perform self-pulse palpation. However, we preferred them to do it at different times of the day. When handheld ECG findings indicated AF or other suspected pathological findings, the ECG was re-examined by an experienced cardiologist in order to confirm the diagnosis. AF was diagnosed as a 30-second recording with an absolutely irregular rhythm without distinct p-waves [1]. Individuals with unclear or uninterpretable ECG recordings were further investigated using patch ECG monitoring for 5 days. The Zenicor device has a button that participants can use to note when their pulse felt irregular during each ECG recording. The responsible author is an experienced family doctor who interpreted the ECG register remotely each day and phoned the participants reminding them to check their pulse and to check whether they felt any AF symptoms when the ECG showed AF. When AF was verified, the patients were contacted by phone, informed about the detected AF, and asked to consult their family doctor in order to validate the indication for AC treatment in accordance with national guidelines. Thus, the patient's family doctor was responsible for the initiation of ACs. A follow-up was performed to assess whether the patient had been prescribed AC treatment. Participants with no detected AF returned the handheld ECG devices at the end of the screening with no scheduled follow-up and were then notified by post about their ECG results.

## Statistical analyses

Categorical data were summarized by counts and percentages. For all continuous variables, visual inspection of histograms and the Shapiro–Wilk's test were used to assess the deviation from a normal distribution. This test showed no normally distributed study data. Thus, medians with interquartile ranges were used. Fisher's exact test was used to analyze categorical variables. Student *t* test or the Mann–Whitney test were used to compare continuous variables between 2 groups. Odds ratios with 95% CIs were used to test for associations between AF and risk factors. Multivariate logistic regression analyses were conducted to analyze the independent predictors for AF detection. An exponential prediction model was used for AF detection by age because this model was most suitable. For these tests, a two-sided probability value $\leq 0.05$ was considered statistically significant. These analyses were performed using Stata statistical software version 10.

The sample size was calculated to show a statistically higher AF detection rate using a 2-week intermittent ECG than an AF detection rate using a single-time–point ECG on inclusion. We assumed a 1.4% AF detection rate using a single-time–point ECG according to a previous meta-analysis [7] for AF screening of patients 65 years of age and older. We expected that total AF detection using a 2-week intermittent ECG would be 3%, depending on a previous study [6] that used this screening method on 76-year–old patients. Using an alpha value of 0.05, power 0.75 to calculate the sample size for the difference in the proportion between 2

dependent groups yielded 955. Thus, we chose a sample size of at least 1,000. We chose 0.75 power because of limited resources.

For pulse palpation for detecting AF, we compared the results of each pulse palpation with the results of a simultaneous ECG recording. We constructed 2 × 2 contingency tables as we had already planned in order to enable calculation of sensitivity, specificity, positive predictive value, and negative predictive value. The planned analyses were not changed during the study, and no data-driven changes of analysis were performed. Calculations with exact Clopper–Pearson CIs were used for sensitivity and specificity, while standard logit CIs were used for the predictive values.

### Ethics

This study was performed in accordance with the Declaration of Helsinki and was approved by the Ethics Committee of Stockholm (DNR 2017/3:3).

## Results

### Participation

Of the 46,477 individuals who were registered at 4 PCCs in 2018, 9,224 (19.85%) were 65 years of age and older. This proportion was similar to the corresponding proportion of the Swedish population (19.9%) but varied across the 4 centers (Table 1).

Screening periods varied from 9 to 15 months in the 4 participating centers. The screening ended at all centers in December 2018 because the prespecified sample size had been included.

Screening was halted during the summer vacation (June and July) at all centers. Furthermore, one of the first 2 centers stopped screening for 3 months because the nurse in charge was on sick leave.

About 90% of registered individuals in the target population visited their PCC during the screening period (Table 2). The prevalence of known AF was 6.79% (95% CI 6.29%–7.32%). Between 4%–28% of all visitors without known AF were screened, with only some of these visitors being invited to be screened. This invitation to the screening depended on the capacity of the research nurse, who carried out her clinical work at the same time. No real estimate of agreement to screening was made, although the recruiting health personal considered the

**Table 1. Participation and AF cases across screening centers.**

| | PCC1 | PCC2 | PCC3 | PCC4 | Total |
|---|---|---|---|---|---|
| Individuals registered at a PCC | 14,901 | 10,797 | 8,344 | 12,435 | **46,477** |
| Number of registered individuals ≥65 years (% of all registered individuals) | 2,987 (20.4) | 2,065 (19.13) | 1,253 (15.02) | 2,919 (23.47) | **9,224 (19.85)** |
| Number of registered individuals ≥65 years who visited the PCC (% of registered individuals in this age group) | 2,739 (91.7) | 1,773 (85.9) | 1,015 (81) | 2,885 (98.8) | **8,412 (91.2)** |
| Number of registered individuals ≥65 years with known AF (% of all registered individuals in this age group) | 216 (7.23) | 194 (9.39) | 111 (8.86) | 105 (3.6) | **626 (6.79)** |
| Number of screened individuals (% of visitors in the target age without previously known AF) | 702 (27.8) | 89 (5.64) | 104 (11.5) | 115 (4.14) | **1,010 (12.97)** |
| Number of new AF cases detected (% of screened individuals) | 18 (2.56) | 1 (1.12) | 4 (3.85) | 4 (3.48) | **27 (2.67)** |
| Median age of screened patients, years | 72.18 | 70.93 | 70.86 | 72.68 | **71.97** |
| Median age of detected AF cases, years | 76.58 | 72.66 | 76.18 | 73.08 | **76.41** |
| Screening period, months | 15 | 15 | 10 | 9 | **9–15** |

**Abbreviations**: AF, atrial fibrillation; PCC, primary care center.

**Table 2. Distribution of target population.**

| | |
|---|---|
| Total number of registered individuals ≥65 years in 4 PCCs | **9,224** |
| Number of registered individuals ≥65 years who did not visit the PCCs (% of all registered individuals ≥65 years) | **812 (8.8)** |
| Number of registered individuals ≥65 years with known AF (% of all registered individuals ≥65 years) | **626 (6.79)** |
| Number of registered individuals ≥65 years without known AF who visited the PCCs (% of all registered individuals ≥65 years) | **7,786 (84.41)** |
| Number of registered individuals ≥65 years without known AF who visited the PCCs but did not participate in the screening (% of registered individuals ≥65 years without known AF who visited the PCCs) | **6,776 (87.03)** |
| Number of screened individuals (% of registered individuals ≥65 years without known AF who visited the PCCs) | **1,010 (12.97)** |

**Abbreviations**: AF, atrial fibrillation; PCC, primary care center.

agreement rate to be around 80%–90%. The median age of the screened individuals was 72.1 years (IQR 68.7, 75.9), of which 61.6% were female.

## AF detection

A total of 1,010 individuals were screened, and 27 (2.7%, 95% CI 1.8%–3.9%) new cases of AF were detected. Only 2 new cases showed persistent AF, while the other cases were paroxysmal. The 2 cases of persistent AF and 3 other cases of paroxysmal AF were detected by the first ECG recording at inclusion. Thus, the rate ratio for AF detection by intermittent ECG compared with single ECG measurement was 5.4 (95% CI 2.3–12.6). Of the 27 new AF cases, 16 (59%) were asymptomatic. Forty-two individuals had nonconclusive ECG recordings, mainly showing a frequent atrial ectopic beat, and were further investigated using BioTelemetry ePatch continuous ECG monitoring (https://www.gobio.com/) for 5 days. AF could be verified in 4 of these individuals.

AC treatment was initiated by the patient's family doctor in 26 of 27 new AF cases (96%, 95% CI 81%–100%). Non-vitamin-K–antagonist oral ACs were initiated in 25 cases, and one other case initially received oral ACs but then changed to low-molecular–weight heparins because of bleeding as a side effect. Only one patient was not treated, as the doctor in charge considered that the patient had a relatively low thromboembolic risk ($CHA_2DS_2$-VASc score = 1).

Table 3 shows the characteristics of newly detected AF cases with a higher age, predominantly male (70.4%) and with more prevalent heart failure. Age and male gender were independent predictors for detection of new AF cases, with an odds ratio (95% CI) of 1.14 (1.07–1.21) and 4.46 (1.9–10.43), respectively. Fig 1 shows the age prediction for detection of AF using an exponential prediction model.

## Validity of pulse palpation

A total of 53,782 simultaneous ECG recordings and pulse measurements were registered for the 1,010 screened individuals, corresponding to a median of 51 recordings/individuals. Of the 27 detected AF cases, AF was verified in 311 ECG recordings (Table 4), but the pulse was palpated as irregular in only 77 of these recordings, yielding a 25% (95% CI 20%–30%) sensitivity and a 98% (95% CI 98%–98%) specificity per measurement occasion (Table 5).

Of these 27 AF cases, 15 cases felt an irregular pulse on at least one occasion, resulting in 56% (95% CI 35%–75%) sensitivity per individual (Table 4). Of individuals without AF, 187

**Table 3. Characteristics of newly detected AF cases compared with those without AF.**

| | AF—27 Patients | No AF—983 Participants | P-Value |
|---|---|---|---|
| Age mean (SD) | 76.8 (7) | 72.7 (5) | **0.0001** |
| Age median (IQR) | 76.4 (72,82) | 72 (69,76) | **0.0016** |
| Female gender number (%) | 8 (29.6) | 614 (62.5) | **0.0005** |
| Heart failure number (%) | 2 (7.4) | 13 (1.3) | **0.0099** |
| Hypertension number (%) | 15 (55.6) | 558 (56.8) | 0.9004 |
| Diabetes mellitus number (%) | 3 (11.1) | 175 (17.8) | 0.3680 |
| Stroke/TIA number (%) | 3 (11.1) | 77 (7.8) | 0.5338 |
| Myocardial infarction/ peripheral artery disease number (%) | 3 (11.1) | 74 (7.5) | 0.4888 |
| $CHA_2DS_2$-VASc score median (IQR) | 3 (2,4) | 3 (2, 4) | 0.9912 |
| Systolic BP median (IQR) | 142 (132, 166) | 140 (129, 152) | 0.1235 |
| Diastolic BP median (IQR) | 83 (77, 92) | 83 (76, 90) | 0.6001 |
| BMI female median (IQR) | 28 (25, 31) | 25 (23, 29) | 0.1939 |
| BMI male median (IQR) | 25 (24, 29) | 26 (24, 28) | 0.5783 |

**Abbreviations**: AF, atrial fibrillation; BMI, body mass index; BP, blood pressure; IQR, interquartile range; SD, standard deviation; TIA, transient ischemic attack.

felt an irregular pulse on at least one occasion, with 81% (95% CI 78%–83%) specificity, and the positive predictive value was 7%.

Pulse palpation by nurse on inclusion was irregular in 25 cases, and 5 new AF cases were detected on inclusion, in which the pulse was irregular in 4 of the cases (Table 4), yielding 80%

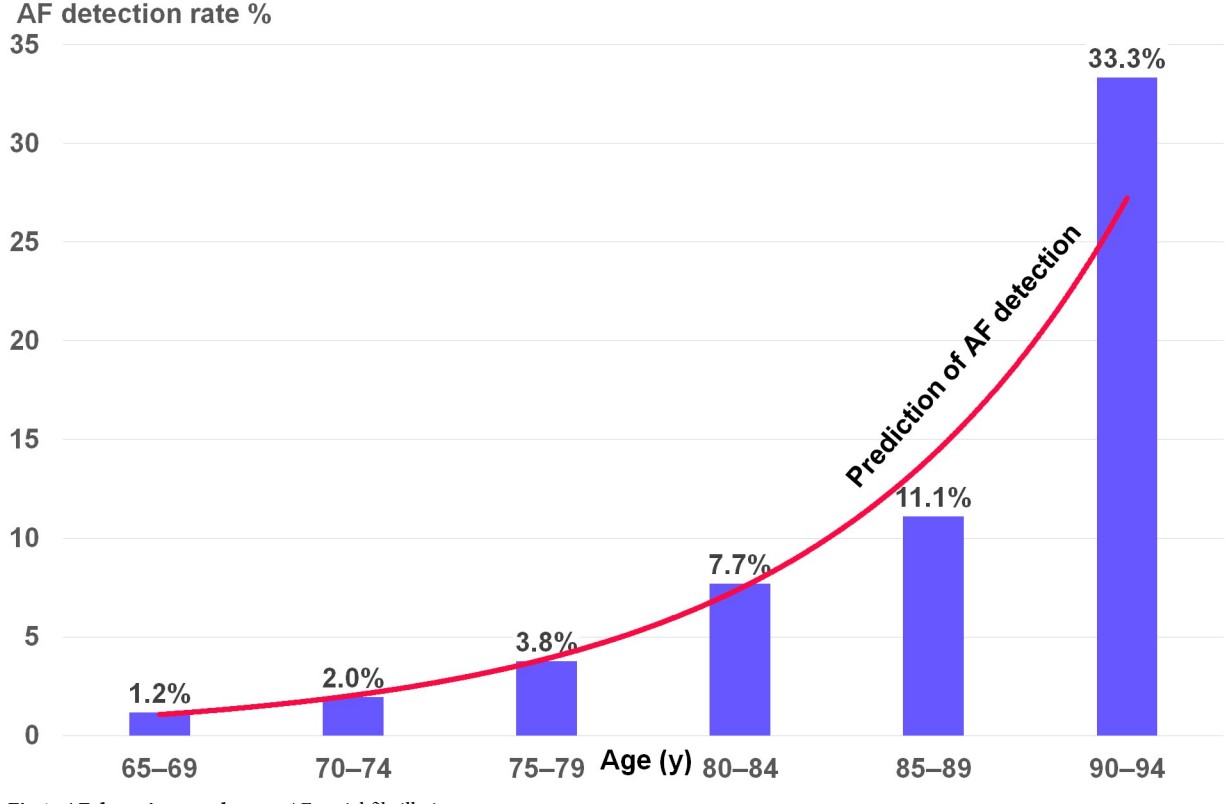

**Fig 1. AF detection rate by age.** AF, atrial fibrillation.

**Table 4. Contingency table comparing pulse results with ECG results.**

| Irregular versus regular pulse | AF—27 Individuals | No AF—983 Individuals |
|---|---|---|
| Irregular pulse—202 individuals | 15 | 187 |
| Regular pulse—808 individuals | 12 | 796 |
| **Pulse measurements** | **AF—311 Measurements** | **No AF—53,471 Measurements** |
| Irregular pulse—1,046 measurements | 77 | 969 |
| Regular pulse—52,736 measurements | 234 | 52,502 |
| **Irregular versus regular pulse** | **AF—5 Individuals** | **No AF—1,005 Individuals** |
| Irregular pulse—25 individuals | 4 | 21 |
| Regular pulse—985 individuals | 1 | 984 |

Abbreviations: AF, atrial fibrillation; ECG, electrocardiogram.

**Table 5. Validity results of pulse palpation in detecting AF.**

| Variable | Pulse Palpation by Nurse on Inclusion versus Single ECG Measurement | | At Least One Irregular Pulse by Individual Self-Pulse Palpation over 2 Weeks versus at Least One ECG Recording with AF | | Individual Self-Pulse Palpation versus Simultaneous ECG | |
|---|---|---|---|---|---|---|
| | **Estimate** | **95% CI** | **Estimate** | **95% CI** | **Estimate** | **95% CI** |
| Sensitivity, % | 80 | 28.36–99.49 | 55.56 | 35.33–74.52 | 24.74 | 20.06–29.94 |
| Specificity, % | 97.91 | 96.82–98.7 | 80.98 | 78.38–83.39 | 98.19 | 98.07–98.3 |
| Positive predictive value, % | 16 | 9.39–25.94 | 7.43 | 5.29–10.32 | 7.36 | 6.09–8.88 |
| Negative predictive value, % | 99.9 | 99.42–99.98 | 98.51 | 97.75–99.02 | 99.56 | 99.53–99.58 |

Abbreviations: AF, atrial fibrillation; ECG, electrocardiogram.

(95% CI 28%–99%) sensitivity, 98% (95% CI 97%–99%) specificity, and 16% (95% CI 5%–36%) positive predictive value (Table 4).

## Discussion

This is the first AF screening study to compare self-pulse palpation with an ECG recording. We found that self-pulse palpation had a low sensitivity and high specificity for AF detection compared with intermittent ECG. Almost all newly detected AF cases were treated with ACs.

Participation appears to depend more on the stability of health personnel and their motivation rather than the motivation of the target population, as only a minority of the care-seeking patients were invited to the screening, although the majority of the target population visited their PCC during the screening period.

Compared with the Swedish general population in the target age (http://www.statistikdatabasen.scb.se/pxweb/en/ssd/), the screened individuals were relatively younger and comprised more females (Fig 2). Thus, we would probably have detected more AF cases if the participants had corresponded more to the Swedish age and gender distribution. The pre-screening prevalence of AF was higher than the AF prevalence observed in other studies [4], probably indicating better routine care with repeated ECG measurements in our studied population.

Our detection rate of AF cases was comparable with a 3% detection rate in a previous screening study [6] that used the same ECG technique among 75- to 76- year–old individuals, although our detection rate was lower compared to a similar study in primary care showing a 5.5% AF detection rate with a median age of 72 years. However, our participants had lower

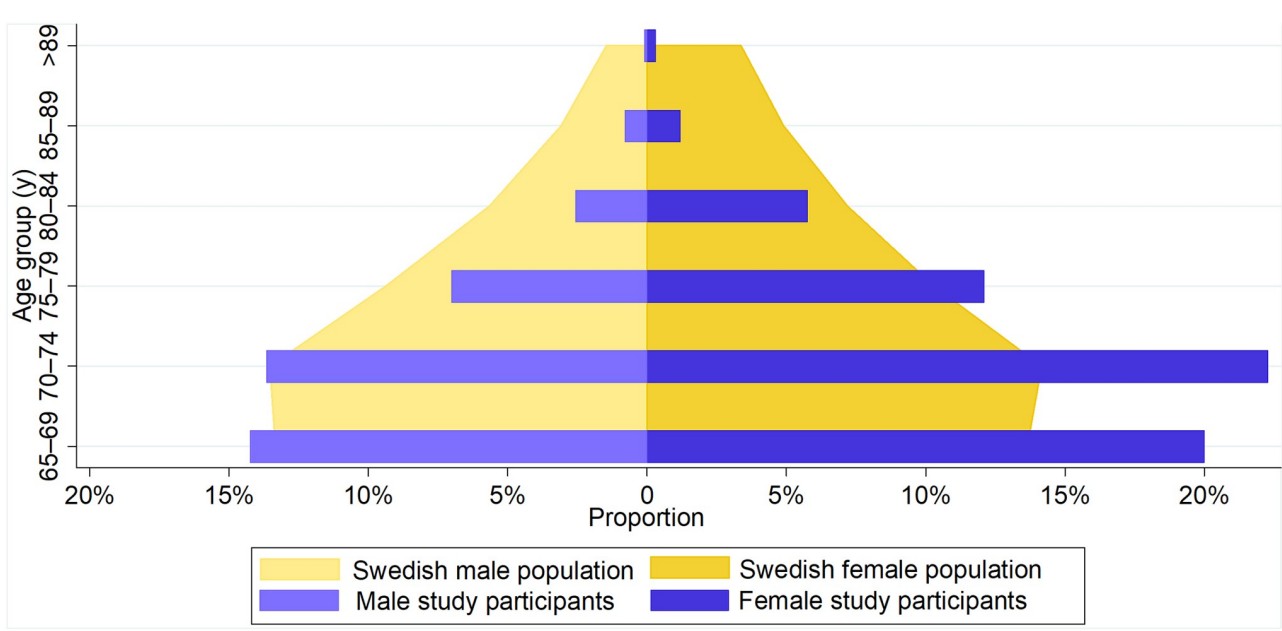

**Fig 2. Age and gender distribution of the study participants compared with the Swedish population.**

morbidities and were predominantly female. In our study, 59% of the detected AF cases were asymptomatic, compared with 75% in the previous screening study [11].

In our study, almost all of the newly detected AF cases were paroxysmal AF. This rate is higher than the paroxysmal AF rates in previous screening studies of 62.5% and 75%, respectively [11, 12]. This may indicate that the majority of nonparoxysmal AF had already been detected via regular healthcare at our screening centers. Thus, in our study, the rate ratio for AF detection by intermittent ECG compared with single ECG measurement was 5.4. Previous screening studies [6, 11] showed similar rate ratios of 4.8 and 5.3, respectively. This indicates the need for prolonged screening in order to detect more new AF cases.

Anticoagulation could be initiated by primary care physicians in almost all newly detected AF cases. This high initiation rate was similar to a previous screening study [11] in primary care.

In our study, age and male gender were independent predictors for the detection of new AF cases. This confirms previous evidence [8] that age and male gender is a strong predictor for AF.

Our analysis that compared self-pulse palpation with simultaneous ECG measurements for AF detection showed low sensitivity. Thus, pulse might be of limited use to AF screening. However, individual analysis based on repeated pulse palpation increases sensitivity when an individual detects an irregular pulse at least once over 2 weeks. Screening tests should be sensitive. A sensitivity of 56% may be not high enough to motivate stepwise screening with self-pulse palpation followed by intermittent ECG recording when the pulse is irregular. Our results showed one AF case detected in 13 individuals with at least one irregular pulse, compared with detecting one AF case in 37 individuals irrespective of pulse palpation.

Our analysis, which was based on a trained nurse checking the pulse as a single measure in the PCC, showed a higher sensitivity to AF detection on inclusion. A meta-analysis [13] of such pulse palpation showed a higher sensitivity of 92% (95% CI 85%–96%). However, it is difficult to detect most paroxysmal AF cases through such single-time measurements. In our study, we detected only 19% of all new AF cases by an initial single ECG recording. Repeated

**Table 6. AF detection by single-time–point measurement pulse palpation and ECG on inclusion versus repeated pulse palpation and ECG measurement over 2 weeks.**

|  | 27 AF Cases Detected by Intermittent ECG (5.4 Times than Those Detected on Inclusion) | 5 AF Cases Detected at First ECG on Inclusion |
|---|---|---|
| Irregular pulse | 15* (56%) | 4** (80%) |
| Regular pulse | 12* | 1** |

*Self-pulse palpation at home.

**Pulse palpation by nurse.

**Abbreviations**: AF, atrial fibrillation; ECG, electrocardiogram.

pulse palpation (even with low sensitivity) detected more AF cases than single-time–point pulse palpation (15 versus 4 cases, as shown in Table 6). Repeated self-pulse palpation at home is inexpensive and easy and requires no special equipment.

A meta-analysis [13] showed the highest sensitivity of single pulse measurement using a smartphone to detect AF of 97% (95% CI 95%–99%), and a recent study showed comparable sensitivity [14, 15]. Again, it is difficult to detect most paroxysmal AF using this single-time–point pulse palpation. Moreover, this screening requires the availability of a smartphone. A handheld smartphone [14, 15] single-lead ECG with an interpreting program can be used for AF screening without the need for pulse evaluation. Thus, if such a smartphone ECG were available, repeated measurements could be a sensitive screening method for AF without pulse checking.

This study has a few limitations. Firstly, PCCs were not randomly selected for recruitment. This could affect the reproducibility of our results. Secondly, a minority of patients who visited PCCs were invited to the screening, and there was no real estimate of participation among the invited patients. The study nurses recruited relatively younger and probably healthier patients. This could cause selection bias. However, the recruitment of patients with higher morbidity would result in a higher AF detection rate. Finally, ectopic heart beats could be felt as an irregular pulse, and this could reduce the specificity of pulse palpation for AF. However, specificity was high in our study.

## Conclusion

AF screening using self-pulse palpation 3 times per day for 2 weeks has lower sensitivity compared with simultaneous intermittent ECG. Using such an ECG is more effective in detecting AF than a single-time–point ECG. Thus, it may be better to screen for AF using intermittent ECG without stepwise screening using pulse palpation. In the future, there is a need to conduct a randomized control screening study for stroke prevention using intermittent ECG.

## Supporting information

**S1 Table. STROBE Checklist.** STROBE, Strengthening the Reporting of Observational Studies in Epidemiology.
(DOCX)

**S2 Table. Data reporting.**
(XLSX)

**S1 Text. Case reporting form.**
(DOCX)

## Acknowledgments

We would like to thank Associate Professor M.D. Johan Engdahl for his help with ECG interpretation. We would also like to thank Elin Westberg from BioTelemetry for her help with interpretation of the ECG loop recorder. Finally, we would like to thank the managers and the nurses for their recruitment of participants at the following PCCs: Aros, Trosa, LäkarGruppen Västerås, and Hälsocentralen City, Gävle.

## Author Contributions

**Conceptualization:** Faris Ghazal, Holger Theobald, Mårten Rosenqvist, Faris Al-Khalili.

**Data curation:** Faris Ghazal.

**Formal analysis:** Faris Ghazal.

**Investigation:** Faris Ghazal.

**Methodology:** Faris Ghazal, Mårten Rosenqvist, Faris Al-Khalili.

**Project administration:** Faris Ghazal.

**Resources:** Faris Ghazal.

**Software:** Faris Ghazal.

**Supervision:** Holger Theobald, Mårten Rosenqvist, Faris Al-Khalili.

**Validation:** Faris Ghazal, Mårten Rosenqvist, Faris Al-Khalili.

**Visualization:** Faris Ghazal.

**Writing – original draft:** Faris Ghazal, Holger Theobald, Mårten Rosenqvist, Faris Al-Khalili.

**Writing – review & editing:** Faris Ghazal, Holger Theobald, Mårten Rosenqvist, Faris Al-Khalili.

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
