## [Decision Letter · Decision Letter 0]

5 Nov 2019

Dear Dr. Ghazal,

Thank you very much for submitting your manuscript "Validity of daily self-pulse palpation over two weeks for screening for atrial fibrillation among patients 65 years of age and older seeking primary care: A cross-sectional study" (PMEDICINE-D-19-03337) for consideration at PLOS Medicine. 

[LINK]

In light of these reviews, I am afraid that we will not be able to accept the manuscript for publication in the journal in its current form, but we would like to consider a revised version that addresses the reviewers' and editors' comments. Obviously we cannot make any decision about publication until we have seen the revised manuscript and your response, and we plan to seek re-review by one or more of the reviewers. 

We expect to receive your revised manuscript by Nov 26 2019 11:59PM. Please email us (plosmedicine@plos.org) if you have any questions or concerns.

We look forward to receiving your revised manuscript. 

Sincerely,

Adya Misra, PhD

Senior Editor 

PLOS Medicine

plosmedicine.org

Title: suggest shortening to “Validity of daily self-pulse palpation for Atrial Fibrillation screening among patients 65 years of age and older: A cross-sectional study”

Abstract: first sentence perhaps should start with “XXX guidelines recommend single time-point screening for atrial fibrillation”. Presumably these are ESC guidelines? 

Abstract methods and findings- please provide a brief description of which primary care centres participated. Perhaps something like “Primary care centres in stockholm”?. Please also provide brief demographics of the participants. Last sentence of this section should include a limitation of your study design

Abstract and author summary- the writing needs to reflect that the study compared the validity of self palpation and handheld ECG recordings. Since you use the word “simultaneously”, it is not clear that these two are being compared. 

Introduction= please introduce AF and AC on first view in Line 78 

Line 79 please rephrase to “may be important to prevent stroke” as there are other risk factors for ischaemic stroke?

Line 84-85 please rephrase to “it is recommended to screen persons 65 years of age and older for AF” for clarity

Please rephrase “head to head” to “direct comparison” or something similar

Introduction- requires further background on atrial fibrillation and the risks of undetected AF. Also provide further detail on self-palpation and whether this is a feasible method to detect AF. Please provide further detail on the 65+ age group and self palpation, along with the various modalities used to detect, diagnose AF.

Methods- Lines 103-104 can be omitted and please provide details of which primary care centres were included. Please also clarify how many PCCs were approached and how many consented to participate for greater transparency 

Methods- please provide a citation to the health questionnaire referred to in Line 125 or provide a copy of this questionnaire as supplementary info. Please also provide a brief summary of what health scores from this questionnaire might mean ( scale of 1 to 5 ? or lower scores are worse than higher?)

Methods- please provide a citation to Zenicor handheld ECG if available and mention which anticoagulants were offered to patients 

Line 173- please clarify the time of the summer vacation during which no screening took place 

Line 197- please mention again 26 of 27 (?) new AF cases?

Fig 1- please remove pie chart and replace with table containing exact numbers and not approximations 

Line 200-201- please provide further detail on why one case was not given treatment and what a score of one means

Fig 2- its not clear how the prediction of AF detection was calculation- please provide this detail in the methods and results sections as necessary 

Please present and organize the Discussion as follows: a short, clear summary of the article's findings; what the study adds to existing research and where and why the results may differ from previous research; strengths and limitations of the study; implications and next steps for research, clinical practice, and/or public policy; one-paragraph conclusion. Please remove all results from the discussion and include these in the results section only.

Please remove assertions of primacy in Line 235-237

Figures 3 is hard to understand, please simplify in line with comments from Ref 1

Figure 4 should be modified into a table

Lines 256-260 require revision for clarity and grammar

Discussion-please discuss in greater detail the potential of remote ECG monitoring in at risk or general populations for AF screening using smartphone apps or other smart devices. 

Discussion- you mention ectopic beats here but there is no context provided previously that the presence of these ectopics can skew the self palpation. Please provide brief summary in the introduction

Conclusion and abstract did this study involve stepwise screening? If so, this needs to be made clearer throughout the text

Data availability statement- you say the data are provided in SI files but we are unable to locate these. Please provide the data underlying tables, figures and charts as per PLOS Data policy

Please provide p values along with confidence intervals as applicable 

Did your study have a prospective protocol or analysis plan? Please state this (either way) early in the Methods section.

c) In either case, changes in the analysis—including those made in response to peer review comments—should be identified as such in the Methods section of the paper, with rationale.

Comments from the reviewers:

Reviewer #1: I confine my remarks to statistical aspects of this paper. The basic approach is fine, but I have some issues to resolve before I can recommend publication.

Line 39 - Should "patients" here be "recordings"?

Line 150 The usual power is 0.80. Why did the authors choose 0.75? (It isn't necessarily wrong to do so, but it needs justification).

Lines 153-155 How were the CIs calculated (there are several methods).

Line 182 Give the interquartile range.

Fig 1 - pie graphs are not good. Here, a simple table would be fine. Also, why were two of the N's approximate?

Fig 3 - this is hard to read. If exact age is available, then it would be better to use two overlaid density plots - one for men and one for women - with lines for screened and general population for each. If age is only available in categories, then two mosaic plots could be presented, or these could be combined into one three-way mosaic plot.

Fig. 4 Stacked bar charts are not good and 3-D makes them worse. I'm not sure what to suggest here. Maybe just a table. Maybe a mosaic plot. Maybe a bar plot in 2D with the bars adjacent to each other.

Peter Flom

Reviewer #2: Ghazal et al have report a cross-sectional screening study in patients >65 years of age visiting primary care centers and taking part in AF screening. Screening was performed with intermittent ECG recordings three times per day for a period of 2 weeks and simultaneous pulse palpation. A total of 1010 patients participated in the study and 27 (2.7%) new cases of AF were detected in 311 ECG recordings, of which the pulse was palpated as irregular in 77 patients (25%). 187 individuals without AF felt an irregular pulse on least one occasion. The specificity per measurement occasion and per individual was (98%). They conclude that AF screening using self-pulse palpation three times daily for two weeks has lower sensitivity compared with simultaneous intermittent ECG. Thus, it may be better to screen for AF using intermittent ECG without stepwise screening using pulse palpation.

The study idea is important, since silent AF is a common problem and stroke is too often its 1st manifestation (PLOS One 2016;11:e0168010). The execution of the study is adequate and the sample size is sufficient for the present study questions. The report is generally well-written.

The weakness of this study is the unclear selection process of study patients weakening the generalizability of the results. 

The sensitivity of pulse palpation was lower than expected showing that this method is not suitable for every elderly subject. In an earlier study, elderly (> 75 years of age) subjects the accuracy of pulse palpation after a training given by study nurse was: sinus rhythm 97%, extra beats 74.3%, slow AF 81.8% and fast AF 91.9% when assessed with the help of anatomic human arm model programmed with various rhythms ( Scand J Prim Health Care 2017;35:93-98). The main problem with pulse palpation is the primary motivation and ability (Parkinson, severe arthrosis in fingers etc) and later the motivation to continue the habit of regular palpation.

Did the study nurses evaluate the ability of subjects to measure their pulse after the training session?

What were the predictors for missing AF with pulse palpation? 

Did the subjects really palpate the pulse and did they report heart rate during each recording?

What was the reason for misdiagnosis of AF?

Reviewer #3: This nicely executed study looks at an important issue, which to my knowledge has not previously been assessed. It aims to determine if self-performed pulse palpation is as accurate at detecting unknown AF as a hand-held ECG, over a period of 2-weeks. This is an important issue to clarify, as the economics of hand-held devices for AF detection are much higher than just educating people to take their own pulse. Overall, this is a nicely presented paper. I believe the methodology is appropriate and the study is powered sufficiently to answer the research question.

There are a few minor things in the paper that I think need to be clarified, and a few places where the English grammar may need a little editing.

1. Line 46-47 - I ma not quite sure that the last sentence of the abstract conclusion means - it reads as if you are suggesting NOT to do step-wise screening, but I do not think that is the case

2. Line 60 - "who searched care" - should this be "who sought care"

3. Line 78 - The initial use of the acronyms AF and AC are not defined on their first use in the body of the manuscript. Additionally, it is more common for OAC to be used rather than AC

4. Line 102 states you recruited in Stockholm county and then line 104 states you screened OUTSIDE Stockholm county

5. Lines 118- 130 - More information is required in the methods section under the screening protocol to describe: how the participants recorded the pulse palpation results; how the ECG and pulse palpation results were assessed to determine AF or no AF; and what other tests were done; how symptoms were determined and recorded; what follow up occurred if AF was identified and how the patient was advised they had AF; who determined the treatment; and how OAC was prescribed

6. Line 147 - there is an updated meta-analysis of screening published by the same authors which may be better to quote instead of reference 4. DOI: 10.1371/journal.pmed.1002903

7. Line 166-167 - Can you explain this a little better - I am not quite sure what this sentence means and it may be a source of bias in the selection of practices that could be worth noting in the limitations

8. I think a further point for the discussion is that the pulse palpation performed by patients themselves is significantly less sensitive and specific than pulse palpation performed by health professionals. There are some previously older studies that have also identified this

[LINK]

---

## [Decision Letter · Decision Letter 1]

2 Dec 2019

Dear Dr. Ghazal,

Thank you very much for submitting your manuscript "Validity of daily self-pulse palpation for atrial fibrillation screening in patients 65 years and older: A cross-sectional study" (PMEDICINE-D-19-03337R1) for consideration at PLOS Medicine. 

[LINK]

We note that the responses to reviewers have not been included within the main text of the manuscript but explained only within the rebuttal letter. Please incorporate the responses to reviewers within the manuscript and ensure that the STROBE checklist has been adhered to as a number of reporting details in the methods appear to be missing. Please see comments from Reviewer 3. Obviously we cannot make any decision about publication until we have seen the revised manuscript and your response, and we plan to seek re-review by one or more of the reviewers. 

We expect to receive your revised manuscript by Dec 12 2019 11:59PM. Please email us (plosmedicine@plos.org) if you have any questions or concerns.

We look forward to receiving your revised manuscript. 

Sincerely,

Adya Misra, PhD

Senior Editor 

PLOS Medicine

plosmedicine.org

Comments from the reviewers:

Reviewer #1: The authors have addressed my concerns and I now recommend publication

Peter Flom

Reviewer #2: I have no further comments. The authors have responded adequately to my previous comments.

Reviewer #3: The authors have done a good job in revising the manuscript, but I note that many comments were not addressed within the manuscript itself - they were just addressed in the response to reviewers. I would suggest that the authors go back through the original reviewer comments and check that each point raised by the reviewers has been answered within the manuscript, unless there is a good reason for why it should not be altered, in which case the response should indicate that nothing was altered because xxx reason.

For example, Reviewer 1 requested that an explanation be provided for the choice of 0.75 statistical power rather than 0.8. This point should be explained in the manuscript, not just to the reviewer.

The methods still require some further clarification - mostly with information that has been brought up in the original reviewer comments - eg:

* how did the nurses identify the eligible patients - were they patients that they were seeing for a clini8cal consultation or did the doctors refer their patients to them

* were the practices offered any incentives for participation, or given anything for the training - did the nurses get accreditation points etc

* did the nurse test/check the patients ability to pulse palpate after training

* when were people asked to do the recordings at home - and how often

* Were they asked to do the pulse and ECG at the same time

* How did they record their pulse palpation result (did they have a diary to write it in?)

* How was the pulse palpation and ECG recordings compared 

* If they had symptoms with the pulse palpation how did they record these

* Who reviewed the ECGs and was this done remotely or with the patient

* Did the patient come back into the centre at the end of the period of time for a final review and to return the device

* The methods should indicate that the family doctor was responsible for prescription of the AC - this is mentioned later in the manuscript, but not in the methods

Limitations

* I don't understand what is meant by the first limitation. It is vague and needs rewording.

* The authors mention in their comments that the different study nurses at each site can introduce a selection bias, but this is not mentioned in the limitations section

The authors state in their response that this was not step-wise screening, but the conclusion of the abstract refers to stepwise screening - this is confusing. Why is step-wise screening introduced as a concept here?

[LINK]

---

## [Editor Report · Decision Letter 2]

29 Jan 2020

Dear Dr. Ghazal,

Thank you very much for re-submitting your manuscript "Validity of daily self-pulse palpation for atrial fibrillation screening in patients 65 years and older: A cross-sectional study" (PMEDICINE-D-19-03337R2) for review by PLOS Medicine.

I have discussed the paper with my colleagues and the academic editor and it was also seen again by reviewers. I am pleased to say that provided the remaining editorial and production issues are dealt with we are planning to accept the paper for publication in the journal.

[LINK]

We look forward to receiving the revised manuscript by Feb 05 2020 11:59PM. 

Sincerely,

Adya Misra, PhD

Senior Editor 

PLOS Medicine

plosmedicine.org

Requests from Editors:

Abstract background: “screening” should be “screening”

Abstract- last sentence of the methods and findings section should be a limitation of your study design.

Conclusion- the last sentence is perhaps overreaching and I would suggest that it is removed

Author summary- could you replace “paroxysmal” with a word that might be more accessible to a non specialist audience?

Please provide a space between text and reference brackets. No additional spaces between the references and full stop are required.

Design section Line 113 should be revised to “ A cross-sectional study was carried out in Swedish primary care centers (PCCs) to screen patients for AF” 

Line 126- “whatever reasons” is not appropriate for a research article. Please rephrase this sentence to “Patients seeking care at a PCC above the age of 65 or older were invited by health personnel to participate in our study”. At Line 136, please mention who explained the study objectives to patients and received informed consent.

Line 182-183, in line with comments from Ref 3, please cite the more recent meta-analysis in addition to the currently cited meta analysis (ref 7) noting that the prevalence is the same in both studies. You may say “The AF detection rate of 1.4% was confirmed by a more recent meta-analysis [ref] which was published after we had completed the planning of this study” or similar.

Line 234- should say “inconclusive” ECG recordings

Line 272-275 could you please clarify in the text whether the irregular pulse noted at inclusion meant that the patients would be excluded from the study at that time, AF verified and AC started? I suspect this information ought to be in the methods too. If the patients were not excluded, please do mention that they carried on self palpating and checking ECG measurements

Discussion- please avoid assertions of primacy and replace with “To our knowledge, self-pulse palpation has not been compared with ECG recordings” 

Line 282-285 appears to be somewhat unnecessary as participation in any form of clinical research does rely on the health personnel recruiting the subjects. I would suggest removing this paragraph. 

Line 289-290 please explain what a Swedish age and gender distribution is, perhaps supported by a reference

The discussion section requires some revision to structure as it jumps from study findings to how these compare with previous studies. Please present and organize the Discussion as follows: a short, clear summary of the article's findings; what the study adds to existing research and where and why the results may differ from previous research; strengths and limitations of the study; implications and next steps for research, clinical practice, and/or public policy; one-paragraph conclusion.

Line 322- I don’t believe stepwise screening has been mentioned anywhere else in accordance with reviewer comments so we recommend this mention is also removed.

Table 6 and any related comments ought to be presented in the results section

Line 349- it is perhaps the generalisability also that is somewhat sacrificed by not randomly selecting various primary care centres in a city or country. It is unclear how you have speculated younger and healthier patients were invited by nurses? Please remove this if there is no reason to believe this might have been the case

You have mentioned ectopic beats in Line 354, 355 but do not mention in the methods if the nurses commented on this while explaining self palpation to patients. Please revise as needed

When resubmitting, please only resubmit the revised version of the manuscript with track changes and a clean version of this. Please do not submit any previous versions of the manuscript. 

Comments from Reviewers:

[LINK]

---

## [Editor Report · Decision Letter 3]

21 Feb 2020

Dear Dr. Ghazal, 

On behalf of my colleagues and the academic editor, Dr. Kazem Rahimi, I am delighted to inform you that your manuscript entitled "Validity of daily self-pulse palpation for atrial fibrillation screening in patients 65 years and older: A cross-sectional study" (PMEDICINE-D-19-03337R3) has been accepted for publication in PLOS Medicine. 

PRODUCTION PROCESS

PRESS

PROFILE INFORMATION

Thank you again for submitting the manuscript to PLOS Medicine. We look forward to publishing it. 

Best wishes, 

Adya Misra, PhD

Senior Editor 

PLOS Medicine

plosmedicine.org